# Tabular data imputation: quality over quantity

## Abstract

Tabular data imputation algorithms allow to estimate missing values and use incomplete numerical datasets. Current imputation methods minimize the error between the unobserved ground truth and the imputed values. We show that this strategy has major drawbacks in the presence of multimodal distributions, and we propose to use a qualitative approach rather than the actual quantitative one. We introduce the kNNxKDE algorithm: a hybrid method using chosen neighbors ($k$NN) for conditional density estimation (KDE) tailored for data imputation. We qualitatively and quantitatively show that our method preserves the original data structure when performing imputation. This work advocates for a careful and reasonable use of statistics and machine learning models by data practitioners.

## 1 Introduction

Big data is often referred to as the "gold of the 21st century". But with ubiquitous large databases, missing data are a pervasive problem. They can introduce a bias, lead to wrong conclusions, or even prevent from using data analysis tools that require complete datasets.

To mitigate this issue, data imputation algorithms have been developed. From the straightforward mean/mode imputation to recent artificial neural networks (ANN) models, a wide range of tools are available to impute incomplete datasets. This study focuses on tabular datasets, i.e. numerical data arranged in rows and columns in a form of a matrix. For tabular datasets, recent benchmarks argue that complex imputation methods do not perform better than simple traditional algorithms [Bertsimas et al., 2018, Poulos and Valle, 2018, Jadhav et al., 2019, Woznica and Biecek, 2020, Jäger et al., 2021]. In particular, the consensus is that the $k$NN-Imputer [Troyanskaya et al., 2001] and MissForest [Stekhoven and Bühlmann, 2012], in spite of being traditional and simple algorithms, generally perform better over a large range of datasets in various missing data scenarios.

Data may be missing because it was not recorded, the record has been lost, degraded, or the data may also be censored. Missing data scenarios are usually classified into three types [Little and Rubin, 2014]: missing completely at random (MCAR), missing at random (MAR) and missing not at random (MNAR). In MCAR the missing data mechanism is assumed independent of the dataset. In MAR, the missing data mechanism is assumed to only dependent on the observed variables. The MNAR scenario encompasses all other possible scenarios: the reason why data is missing may depend on the missing value itself. Most comparisons focus on the MCAR scenario.

Tabular data imputation methods have always been evaluated using the RMSE between the estimated value and the ground truth. The higher the mean RMSE, the poorest the imputation method. This approach is of course intuitive, but is too restrictive for multimodal datasets: it assumes that for a set of observed variables, there exists only a unique answer to recover. For multimodal datasets, density estimation methods like the familiar Kernel Density Estimation (KDE) [Rosenblatt, 1956, Parzen, 1962], appear of interest for data imputation. But despite some attempts [Titterington and Mill, 1983,

Leibrandt and Günnemann, 2018], density estimation methods do not handle well observations with missing values.

In this paper, we propose to step back and look at simple datasets to demonstrate that current approaches for data imputation have serious shortcomings. To tackle them, we introduce a local density estimator tailored for data imputation. By leveraging the convenient properties of the $k$NN-Imputer and KDE, we develop kNNxKDE: a simple yet efficient algorithm for stochastic local data imputation. We visually show that our method performs better than standard methods, and evaluate the performances using the likelihood when available. We provide the code and the data used in this work for reproducibility. Interested readers may experiment with the hyperparameters of our algorithm.

## 2 Current methods perform poorly for multimodal dataset

This section demonstrates that conventional data imputation methods provide poor imputation with basic multimodal datasets. For this purpose, we generate three simple two-dimensional datasets and visually assess the imputation performances of four standard methods.

### 2.1 Three simple datasets

The first dataset is a bijection. $x_1$ is sampled from a mollified uniform distribution on $[0, 1]$ with standard deviation $\sigma = 0.05$. Then $x_2 = x_1 + \varepsilon$, where $\varepsilon \sim N(0, 0.1)$.

The second dataset is a surjection, using a sine wave: $x_1 = 4\pi u$, where $u$ is sampled from a mollified distribution on $[0, 1]$ with standard deviation $\sigma = 0.05$. Then $x_2 = \sin x_1 + \varepsilon$, where $\varepsilon \sim N(0, 0.2)$. The surjection allows to show that most imputation algorithms perform well in the unambiguous case (when $x_2$ is missing), but not with multimodal distributions (when $x_1$ is missing).

Finally, Dataset 3 displays a ring. It has been generated in polar coordinates: $\theta \sim \mathcal{U}[0, 2\pi]$ and $r = 1.0 + \varepsilon$, where $\varepsilon \sim N(0, 0.1)$. Euclidean coordinates are $x_1 = r \cos \theta$ and $x_2 = r \sin \theta$.

All three datasets have $N = 500$ observations and are plotted in Figure 1. The code used for generation and the datasets themselves are provided in supplementary materials. We have used a mollified uniform distribution for $x_1$ in Datasets 1 and 2 to prevent from zero likelihood computation problems at the edges of the uniform distribution.

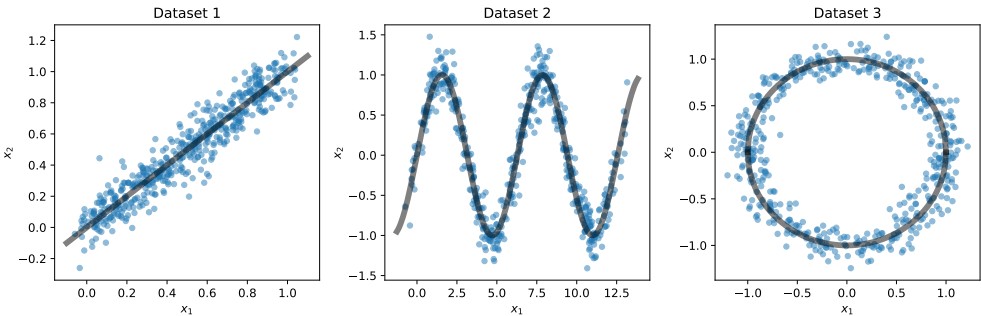

Figure 1: Three basic synthetic datasets with $N = 500$ observations. Dataset 1 is a bijection, Dataset 2 is a surjection, and Dataset 3 uses polar coordinates (not a function in the euclidean space).

### 2.2 Four standard data imputation methods

Here, we present the four data imputation methods used in this work: the $k$NN-Imputer, MissForest, MICE and GAIN. This choice is of course arbitrary, but illustrates well the current state of affairs regarding tabular data imputation [Bertsimas et al., 2018, Poulos and Valle, 2018, Yoon et al., 2018, Jadhav et al., 2019, Woznica and Biecek, 2020, Jäger et al., 2021]

- The $k$NN-Imputer [Troyanskaya et al., 2001] computes distances between pairs of observations using a Euclidean distance that can handle missing values (called nan-Euclidean

distance). It imputes missing values by looking at one column at a time and averaging over the $k$ nearest neighbors that have an observed value for that column. Therefore, different neighbors can be used to impute two missing entries in the same observation. One needs to tune the hyperparameter $k$ for the number of neighbors. The scientific consensus puts the $k$NN-Imputer often on par with MissForest as for the best tabular data imputation method.

- MissForest [Stekhoven and Bühlmann, 2012] is an iterative imputation algorithm. It begins by filling all missing values with initial estimates (e.g. the column mean), and then loops through all columns, one at a time, performing a regression of that specific column onto all other columns using Random Forests. It stops when the imputed dataset is stable enough (following a user-defined threshold). The number of trees has to be tuned. MissForest has shown great flexibility and successful data imputation results.

- MICE stands for Multiple Imputation Chained Equations [van Buuren and Groothuis-Oudshoorn, 2011]. Similar to MissForest, it is an iterative imputation algorithm that uses a regressor (linear regressions for MICE) to predict each column successively after filling all missing entries with initial guesses. This algorithm has no hyperparameter to optimize. MICE has shown good imputation results and is appreciated for its simplicity and absence of hyperparameter tuning, but it fails at capturing non-linear dependencies.

- Finally, GAIN is a GAN neural network tailored for tabular data imputation which claims state-of-the-art imputation results [Yoon et al., 2018]. GAIN smartly revisits the GAN architecture by working with individual cells rather than whole observations. It has benefited from a lot of attention for tabular data imputation. However, recent benchmarks show that its performances are mediocre in practice [Jäger et al., 2021]. GAIN has several hyperparameters to tune: batch size, hint rate (amount of correct labels provided to the discriminator), number of training iterations, and weight parameter $\alpha$ for the generator loss (balances RMSE loss for the observed cells and adversarial loss for the generated cells). We decide to follow the authors' recommendations and fix: batch size $N_{\text{batch}} = 128$, hint rate $r_{\text{h}} = 0.9$ and $\alpha = 100$. We only optimize the number of iterations.

## 2.3 Imputation results

We introduce missing values for each dataset in a MCAR scenario with 20% missing rate. If an observation has both features removed, we repeat the process until at least one feature is present. After missing values have by injected, we normalize the dataset in the range $[0, 1]$ using the minimum and maximum value of each feature.

For each data imputation algorithm and for each dataset, we perform a grid search of the hyperparameter than best minimizes the normalized RMSE (NRMSE):

$$\text{NRMSE} = \sqrt{\frac{1}{N_{\text{miss}}} \sum_{i=1}^{n} \sum_{j=1}^{d} (x_{ij} - \widehat{x}_{ij})^2 m_{ij}}$$

where $m_{ij} = 1$ if cell $(i, j)$ is missing ($m_{ij} = 0$ otherwise) and $N_{\text{miss}} = \sum_{i=1}^{n} \sum_{j=1}^{d} m_{ij}$ is the total number of missing entries in the dataset. The best hyperparameters, presented in Table 1, are used to impute each dataset one more time. The optimized imputation results are plotted in Figure 2.

Table 1: Hyperparameter search results for each imputation method and dataset

| | Data imputation method | | | |
|---|---|---|---|---|
| | $k$NN-Imputer | MissForest | MICE | GAIN |
| Dataset 1 | $k = 30$ neighbors | $N_{\text{trees}} = 10$ | X | $N_{\text{iter}} = 500$ |
| Dataset 2 | $k = 30$ neighbors | $N_{\text{trees}} = 30$ | X | $N_{\text{iter}} = 200$ |
| Dataset 3 | $k = 75$ neighbors | $N_{\text{trees}} = 30$ | X | $N_{\text{iter}} = 100$ |

We believe that Figure 2 provides meaningful insight regarding the current state of tabular data imputation. The scientific consensus is that the $k$NN-Imputer and MissForest provide overall better data imputation quality, which is somewhat recovered here. MICE uses linear regression between features and cannot capture non-linear dependencies. Despite its flexible architecture, GAIN do not recover missing values, even for Dataset 1. GAIN is hard to train properly.

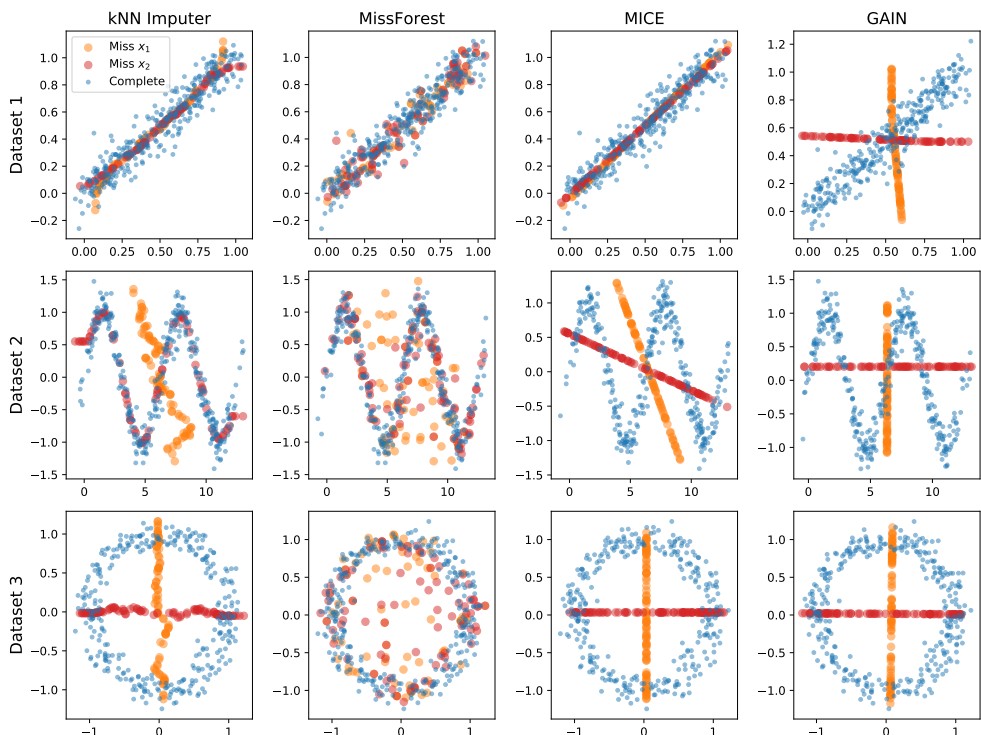

Figure 2: Imputation results for the three synthetic datasets by the four selected imputation methods with optimized hyperparameters. Blue dots correspond to complete observations, orange dots have observed $x_2$ but imputed $x_1$, and red dots have observed $x_1$ but imputed $x_2$. The $k$NN-Imputer, MissForest and MICE perform well on Dataset 1. The $k$NN-Imputer and MissForest can impute $x_2$ for Dataset 2, but they cannot impute $x_1$. No method can properly impute Dataset 3. GAIN provides the worst imputation results and cannot even impute Dataset 1.

Both the $k$NN-Imputer and MissForest average over several predictions. This is why the imputation of $x_1$ in Dataset 2 lies between the two sine waves, and imputations for both $x_1$ and $x_2$ in Dataset 3 are inside the ring. While averaging over several predictions often lead to better estimates, this strategy deteriorates the imputation quality if the missing value distribution is not unimodal.

MICE performs imputation by assuming linear dependency between features in the dataset. It is therefore no surprise if MICE can very well impute Dataset 1 but fails at imputing Dataset 2 and Dataset 3. Once the MICE algorithm has converged, the imputed orange and red dots follow almost perfectly the center of mass of all points in the dataset.

GAIN provides surprisingly disappointing imputation results. While ANNs are flexible models, the generator and the discriminator of GAIN fail to capture the non-linear relationship between $x_1$ and $x_2$ in all three datasets. Because of its innovative and complex framework, GAIN suffers from a complicated training process, which leads to bad imputation results. We have tried to train GAIN several times with various hyperparameters, but always end up with similar imputation quality.

## 3   kNNxKDE

To address the issues presented in Section 2, we propose a local stochastic imputer using kernel density estimation with Gaussian kernels. We adapt the KDE algorithm to missing data settings: only the conditional density of missing features given the observed features is estimated.

We use a methodology analogous to the $k$NN-Imputer to look for neighbors, but we work with missing patterns instead of working column by column. The reason of this choice is that working with one column at a time may lead to incoherent imputations as the selected neighbors for different

columns are different. Therefore, some imputed observations may be incompatible with the dataset structure. For a dataset with $D$ columns, we have up to $2^D - 2$ possible missing patterns. Indeed, each cell may either be missing or not (hence $2^D$ choices) but we do not account for complete cases (nothing to impute) and completely unobserved cases (without even an observed cell).

For each pair of observations in the normalized dataset, we compute the distance $d_{ij}$ using the nan-Euclidean distance [Dixon, 1979]:

$$d_{ij} = \sqrt{\frac{D}{|\mathcal{D}_{\text{obs}}|} \sum_{k \in \mathcal{D}_{\text{obs}}} (x_{ik} - x_{jk})^2}$$

where $D$ is the total number of columns in the dataset, $\mathcal{D}_{\text{obs}} = \{k \in [\![1, D]\!] \mid m_{ik} = m_{jk} = 1\}$ is the set of indices for commonly observed features in observations $i$ and $j$ and $|\mathcal{D}_{\text{obs}}|$ is its cardinality. These pairwise distances are then passed to a softmax function in order to define probabilities:

$$p_{ij} = \frac{e^{-\tau d_{ij}}}{\sum_j e^{-\tau d_{ij}}}$$

We use the "soft" version of the $k$NN algorithm, and introduce the temperature hyperparameter $\tau$. Instead of selecting a fixed number of neighbors per observation, we use a neighborhood where nearest neighbors have stronger weights. In a similar fashion as Frosst et al. [2019], the notion of temperature controls the tightness of each observation's neighborhood.

Given a missing pattern, we first select all data to impute and potential donors. Data to impute is the subset of data which has the current missing pattern, and potential donors are the subset of data where at least all columns in the current missing pattern are observed. For an incomplete observation $i$ in the subset of data to impute, $p_{ij}$ is the probability of choosing observation $j$ from the subset of potential donors. We have $\sum_j p_{ij} = 1$. Algorithm 1 shows the pseudo-code of the kNNxKDE.

The kNNxKDE has three hyperparameters. The temperature $\tau$ for the softmax probabilities, the (shared) standard deviation $h$ of the Gaussian kernels, and the number $N_{\text{draws}}$ of total sampled neighbors. The temperature $\tau$ controls the breadth of the selected neighborhood. The standard deviation $h$ corresponds to the width of the Gaussian kernels. The effects of $\tau$ and $h$ are discussed in Section 4. The last hyperparameter is the number $N_{\text{draws}}$ of imputation samples to be returned. It determines the resolution of the estimated density. Besides the obvious computational resources, there are no drawbacks to setting a high number of imputation samples $N_{\text{draws}}$.

---
**Algorithm 1:** Pseudo-code for the kNNxKDE
___
**Data:** The incomplete dataset $X$
min/max normalization;
**for** *each missing pattern* **do**
    $X_{\text{imp}} \leftarrow$ `data_to_impute`;
    $X_{\text{don}} \leftarrow$ `potential_donors`;
    $d_{ij} \leftarrow$ `nanEuclDist` $(X_{\text{imp}}, X_{\text{don}})$;
    **if** $d_{ij}$ *is `NaN`* **then**
        | $d_{ij} \leftarrow \infty$;
    **end**
    $p_{ij} \leftarrow$ `softmax` $(-\tau d_{ij})$;
    **for** *each row in* $X_{\text{imp}}$ **do**
        $r \leftarrow$ sample $N_{\text{draws}}$ indices in $X_{\text{don}}$ with prob $p_{ij}$;
        $e \leftarrow$ sample $N_{\text{draws}}$ from $e \sim \mathcal{N}(0, h)$;
        `imputation_samples` $\leftarrow X_{\text{don}}[r] + e$;
    **end**
**end**
min/max renormalization;
**Return:** `imputations_samples`

---

## 4   Results on synthetic datasets

In Subsection 4.1, we show the performances of the kNNxKDE on the three artificial datasets and we discuss the effect of the hyperparameters $\tau$ and $h$. In Subsection 4.2, we use the log-likelihood of the imputed sample as an attempt to quantify imputation quality. We show that, for multimodal datasets, using the likelihood is more appropriate than the RMSE. All experiments use the MCAR setting to artificially introduce missing data with 20% missing rate.

## 4.1 Qualitative evaluation of the kNNxKDE algorithm

We show that the proposed method provides imputation samples that preserve the structure of the original dataset. For now, we fix the hyperparameters of the kNNxKDE at their default values: $h = 0.03$, $\tau = 50.0$ and $N_{\text{draws}} = 10000$. Figure 3 shows the imputation with a sub-sampling size $N_{\text{ss}} = 10$. The sub-sampling size is only used to show the variability in the imputation results by sampling several times. If $x_1$ is missing, we sample $N_{\text{ss}}$ possible values given $x_2$ (the orange horizontal trails of dots), and if $x_2$ is missing, we draw $N_{\text{ss}}$ possible estimates given $x_1$ (the red vertical trails of dots).

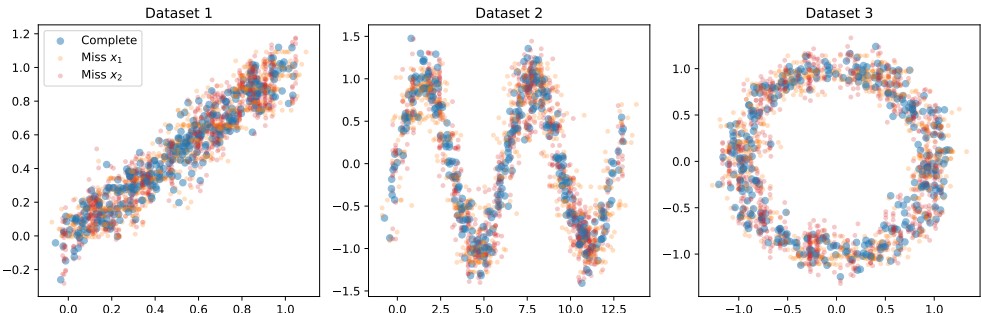

Figure 3: Several imputation results from the kNNxKDE algorithm. Each missing entry has been imputed $N_{\text{ss}} = 10$ times to show the variability of the estimates. The imputed values match with the structure of the observed data (larger blue dots).

Another way to visualize the distribution of the conditional distribution for each missing value is to look at the univariate density provided by the kNNxKDE algorithm. For each dataset, we have selected two observations: one with missing $x_1$ and one with missing $x_2$. Figure 4 shows six univariate densities returned by the kNNxKDE algorithm with default hyperparameters values. In the upper left corner of each panel, the observed value is shown for reference. On each panel, a thick dashed line indicates the (unknown) ground truth. We see that the ground truth always falls in one of the modes of the estimated imputation density.

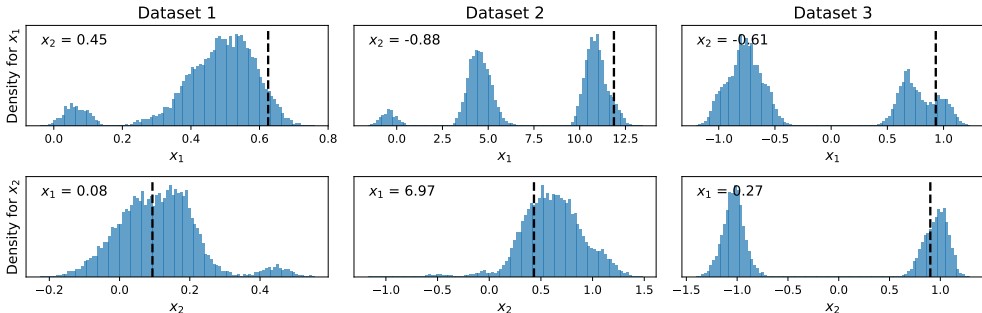

Figure 4: Example of conditional density distributions from the kNNxKDE algorithm with default hyperparameter values. Each histogram has $N_{\text{draws}} = 10000$ samples. Thick dashed lines correspond to the (unobserved) ground truth and the observed value is in the upper-left corner.

For Dataset 2, when $x_1$ is missing (upper middle panel of Figure 4), the kNNxKDE returns a multimodal distribution. Indeed, given the observed $x_2 = -0.88$, three separate ranges of values could correspond to the missing $x_1$. Similarly, Dataset 3 shows bimodal distributions both for $x_1$ or $x_2$, corresponding to the two possible ranges of values allowed by the ring structure.

We now focus on Dataset 2 to experiment with the hyperparameters $h$ and $\tau$. Figure 5 shows how the imputation quality changes when we vary the softmax temperature $\tau$, and the effects of the Gaussian kernel bandwidth $h$ are shown in Figure 6.

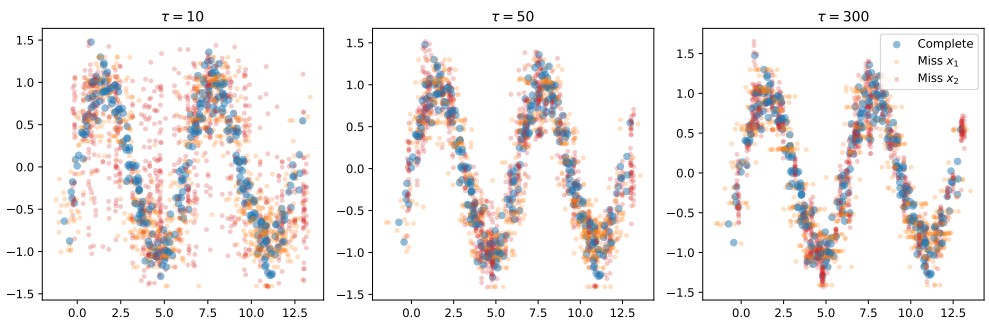

Figure 5: Evolution of the imputation quality as the softmax temperature $\tau$ varies. The Gaussian kernel bandwidth is fixed at h=0.03. We see that if $\tau$ is too low, the imputation has a large variance. If $\tau$ is too high, the imputation could be biased.

The value of the softmax temperature $\tau$ plays an important role in the data imputation quality, as can be seen in Figure 5. Recall that $\tau$ constrains the neighborhood range for each observation. The lower $\tau$, the looser the neighborhood, and irrelevant observations could be sampled. This results in a large scatter (leftmost panel). Conversely, the higher $\tau$, the tighter the neighborhood. Missing values will be imputed using very few other observations and multimodality can be overlooked. This can be seen on the rightmost panel, where the sampling variability is only due to the Gaussian kernel bandwidth. Tuning $\tau$ means finding a good balance in the bias/variance tradeoff.

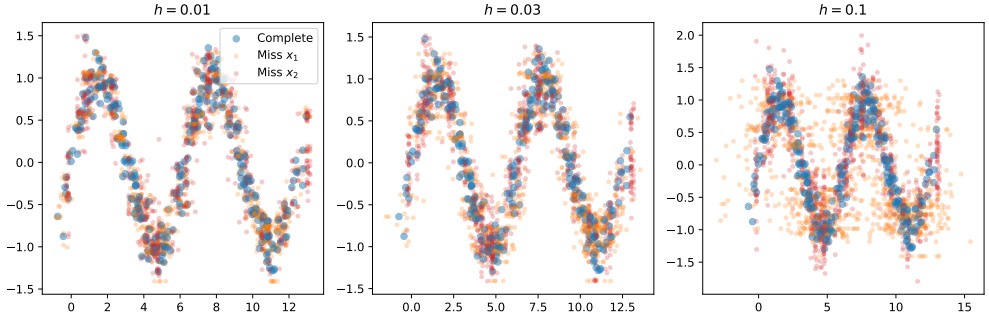

Figure 6: Change in the imputation quality when the Gaussian kernel bandwidth $h$ varies. The softmax temperature is fixed at $\tau = 50$. We see that if $h$ is too low, the imputation sample is very close to the observed data. If $h$ is too high, the imputation sample is too scattered.

Now, the kernel bandwidth $h$ controls the amount of fit to the observed data (c.f. Figure 6). The lower $h$, and the closer to the observed data the imputation sample will be. This can result in spiky univariate distributions. In the limit where $h = 0.0$, the conditional distribution for each missing value becomes a multinomial distribution with probability given by the softmax function computed with the pairwise distances. On the contrary, the higher $h$ and the higher the variability of the imputation sample. Unlike $\tau$, a bandwidth $h$ too narrow does not mean that multimodality will be overlooked. With low $h$, the univariate distribution for a multimodal conditional probability will show distinct pronounced peaks. If $h$ is too high, the different modes may collapse into a larger distribution with high variance.

## 4.2 The log-likelihood to measure imputation quality

Here, we compute the normalized RMSE (NRMSE) for the three datasets after imputation with all standard methods and the kNNxKDE algorithm. We compare the NRMSE with the log-likelihood score, which we can also compute since we know the generative process of the synthetic datasets. When performing a single imputation with the kNNxKDE algorithm, we draw a unique random sample from the resulting imputation distribution.

For each dataset and each imputation method, we repeat 100 times the following process: we introduce
missing values, normalize the dataset, impute with the selected method using best hyperparameters
(c.f. Table 1) and compute the NRMSE. Table 2 shows the mean and the standard deviation of the
NRMSE. As already discussed in Section 2, the $k$NN-Imputer, MissForest and MICE have a low
RMSE for Dataset 1, meaning that these methods recover well missing values. Larger NRMSEs for
Datasets 2 and 3 quantify the poorer imputation quality. GAIN has a large RMSE, even for Dataset 1,
as it could be anticipated from Section 2.

Table 2: Normalized RMSE for the three datasets with all imputation methods. kNNxKDE does not
perform particularly well in terms of minimizing the NRMSE.

| | Data imputation method | | | | |
|---|---|---|---|---|---|
| | $k$NN-Imputer | MissForest | MICE | GAIN | kNNxKDE |
| Dataset 1 | $0.075 \pm 0.005$ | $0.096 \pm 0.005$ | $0.075 \pm 0.004$ | $0.228 \pm 0.026$ | $0.111 \pm 0.006$ |
| Dataset 2 | $0.192 \pm 0.011$ | $0.252 \pm 0.019$ | $0.250 \pm 0.009$ | $0.271 \pm 0.023$ | $0.267 \pm 0.017$ |
| Dataset 3 | $0.295 \pm 0.010$ | $0.374 \pm 0.022$ | $0.294 \pm 0.010$ | $0.309 \pm 0.027$ | $0.419 \pm 0.024$ |

The kNNxKDE does not perform well with the RMSE. It has the largest NRMSEs, if we disregard
GAIN. The justification we provide is that the kNNxKDE is not designed to accurately recover
missing values. When performing a single imputation, the kNNxKDE algorithm selects a unique
sample from the resulting imputation distribution. This is equivalent to selecting a single neighbor
with the softmax probabilities – which may not even be the closest neighbor – and using a noisy copy
of its observed values for imputation. This is an audacious choice, while the other imputation methods
look for an optimal compromise. For multimodal distributions, sampling with the kNNxKDE cannot
guarantee that we sample from the mode where the ground truth lies. For Dataset 3, where kNNxKDE
shows the highest NRMSE, the imputation may be completely off (i.e., on the other side of the ring).

We now compute the log-likelihood of the resulting imputed sample. Like with the NRMSE, for
each dataset and each imputation method, we repeat 100 independent experiments with the best
hyperparameters. The imputed data are renormalized back to their original range to compute the
log-likelihood of the imputed samples. Table 3 shows the mean and the standard deviation of the
log-likelihood.

Table 3: Mean and standard deviation of the log-likelihood for the three datasets with all imputation
methods. The first column shows the log-likelihood of the original sample for reference.

| | **Ref.** | Data imputation method | | | | |
|---|---|---|---|---|---|---|
| | | $k$NN-Imputer | MissForest | MICE | GAIN | kNNxKDE |
| Dataset 1 | 425 | $494 \pm 9$ | $450 \pm 14$ | $495 \pm 11$ | $-234 \pm 231$ | $408 \pm 15$ |
| Dataset 2 | 79 | $-2214 \pm 299$ | $-525 \pm 150$ | $-2691 \pm 261$ | $-1482 \pm 600$ | $-54 \pm 33$ |
| Dataset 3 | -481 | $-2251 \pm 196$ | $-893 \pm 117$ | $-2361 \pm 209$ | $-2117 \pm 319$ | $-509 \pm 15$ |

This time, kNNxKDE performs best for Datasets 2 and 3. For Dataset 1, the $k$NN-Imputer, MissForest
and MICE have a larger log-likelihood than the original sample because these methods average over
several predictions and therefore remove the variability in their predictions: the imputed sample is
very close to the ground truth and shows a high likelihood under the generative model (c.f. Figure 2).
The log-likelihood of the imputed samples by GAIN is poor regardless of the dataset. MissForest
shows interestingly decent results as it benefits from the iterative imputation mechanism and the
random forest flexibility to capture non-linear dependency (unlike MICE).

With the log-likelihood as the new evaluation metric, the kNNxKDE now provides the best imputed
samples. Each imputed observation may be far from its ground truth – hence the large NRMSE in
Table 2, but it conforms to the data structure – hence the large log-likelihood in Table 3.

## 5  Discussion

We have shown the limits of the RMSE for data imputation problems, and have introduced a new
data imputation method. In this last section, we talk about the limitations and the strengths of the

kNNxKDE algorithm, and summarize the main findings. We also provide recommendations for data scientists and statisticians, be it for industry, research or public organizations.

## 5.1 Limits

The obvious major drawback of the kNNxKDE is that we do not provide a clear way to optimize it. We showed that our method performs best in terms of likelihood, but real-world datasets do not come with a likelihood. Therefore, we are left with two options: either we use visual inspection and plots to assess the data imputation quality, or we optimize $\tau$ to minimizing the RMSE (c.f. Appendix A).

Also, the kNNxKDE algorithm may not be suited for highly dimensional datasets. Not only can it become computationally expensive, but its performances shall also worsen. Indeed, because of the curse of dimensionality, initially close observations may end up far apart if similar features are unobserved. This effect becomes even more problematic in high missing rates settings: as we work with missing rate patterns, observations with few observed features will have a small number of potential donors. This problem can be mitigated if the dataset has many observations. As a consequence, calibrating the kNNxKDE algorithm in high dimensions is particularly challenging. Pairplots may be used to visually assess the imputation quality, but become inconvenient in high-dimension settings. Also, pairplots only display pairwise correlations and may overlook higher order structures (c.f. Appendix B).

## 5.2 Strengths

If minimizing the imputation RMSE is an intuitive strategy for tabular data imputation, it cannot capture the complexity of multimodal datasets. In practice, given an incomplete observation, if two different imputations are consistent with the rest of the observed dataset, we have no objective way of choosing one over the other. The kNNxKDE offers to not choose between these two options instead of averaging over them both. It returns a imputation sample that provides more information that a single point estimate.

Unlike the $k$NN-Imputer which impute column after column, the kNNxKDE works with successive missing patterns. This allows to generate imputed samples which are consistent with the whole dataset. Since all missing features are imputed at the same time, this strategy cannot return anomalous imputed samples.

## 5.3 Conclusion

The main motivation of this work was to design an algorithm capable of imputing missing features of a dataset with several modes. Multimodality makes imputation ambiguous, as clearly distinct values may still be valid imputations. In this respect, we decide to use the likelihood as a metric of imputation quality, instead of the standard RMSE between ground truth and imputed samples. The kNNxKDE method does not aggregate estimations. Instead, it returns imputation samples all consistent with the observed dataset. If needed, minimizing the imputation RMSE is possible by averaging over the imputation samples, although we discourage from straightforwardly doing so as it may lead to inconsistent imputed observations (c.f. Appendix A).

Ultimately, this work advocates for a qualitative approach of data imputation, rather than the current quantitative one. We believe that missing data imputation should be done carefully and meaningfully, as it influences subsequent data analysis. We provide the kNNxKDE algorithm, and we suggest trying it for practical tabular data imputation in various domains.

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

## A    Real-world dataset: minimizing the RMSE with kNNxKDE

For practical purposes, one may remain interested in minimizing the RMSE between the imputed
sample and the ground truth. This appendix shows how to use the kNNxKDE to obtain similar
RMSE performances as standard data imputation methods. The imputation samples returned by
the kNNxKDE allow for many ways of performing a single imputation. Rather that sampling the
conditional distributions only once for imputation – like we did in Section 4 – we can compute
appropriate statistics to estimate the missing values. Here, we use the mean for imputation.

The hyperparameter $\tau$ of the kNNxKDE is tuned to minimize the imputation NRMSE when using
the mean for the imputation. We use the Penguins dataset [Horst et al., 2020]: 342 penguins with 4
features (beak length, beak depth, flipper length and body mass) organized in 3 classes. This dataset

Table 4: Mean and standard deviation of the NRMSE on the Penguins dataset with all imputation methods. Optimal hyperparameters (shown below each method name) are obtained to minimize the NRMSE. kNNxKDE(m) stands for imputation performed with the mean of the returned samples from the kNNxKDE.

| $k$NN-Imputer 40 neighbors | MissForest 30 trees | MICE x | GAIN 1200 iterations | kNNxKDE default | kNNxKDE(m) $\tau = 15$ |
|---|---|---|---|---|---|
| $0.136 \pm 0.008$ | $0.147 \pm 0.012$ | $0.154 \pm 0.008$ | $0.186 \pm 0.026$ | $0.219 \pm 0.014$ | $0.140 \pm 0.012$ |

is similar to the famous `iris` dataset. Results are reported in Table 4, where hyperparameters are optimized to minimize the NRMSE.

As we can see, averaging over the conditional distributions leads to similar performances as with the standard $k$NN-Imputer. The difference is that we now tune the continuous hyperparameter $\tau$, which defines how loose the neighborhood of each observation is, rather than the number of neighbors $k$ for the standard $k$NN-Imputer.

Note that, while the resulting imputation minimizes the RMSE, this may not preserve the structure of the original dataset any longer. If the original dataset is multimodal, the imputed dataset can present inconsistent observations.

# B Synthetic data in 3d: visualizing higher-order correlations

We generate a dataset in 3-dimensions using spherical coordinates. Pairplots cannot help visualizing beyond pairwise correlations. But some structures may involve higher-order dependencies which traditional data imputation algorithms do not capture. For example, Figure 7 compares the imputation of the 3-d synthetic dataset with the $k$NN-Imputer and with the kNNxKDE. Table 5 presents the NRMSE and the log-likelihood for each method.

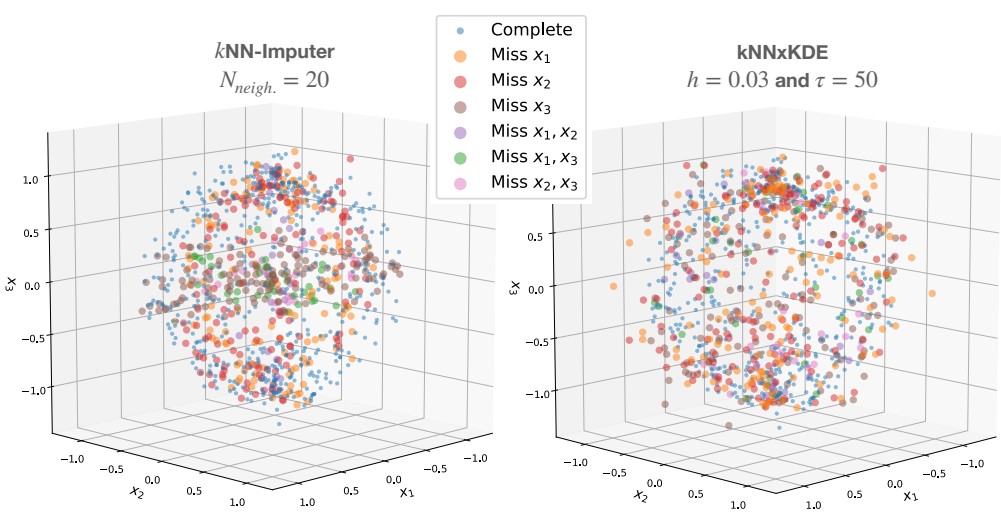

Figure 7: Visualization of the imputed 3-d spherical dataset (MCAR scenario with 20% missing rate): $k$NN-Imputer (left panel) and kNNxKDE (right panel). Points colors indicate imputed components. The $k$NN-Imputer creates artifacts (points inside the sphere) while the kNNxKDE preserve the original dataset structure.

Regarding the NRMSE, the kNNxKDE performs bad. But using the log-likelihood as benchmark, we see that the random sample generated by the kNNxKDE is much more probable under the generative model, i.e. the imputed sample is consistent with the original dataset. The scatter of the imputed observations (right panel of Figure 7) can be adjusted with $\tau$ and $h$.

Visual animations of the imputed samples with all five imputation methods are provided as supplementary materials, where we can notice the characteristics of each imputation method.

Table 5: Mean and standard deviation of the NRMSE on the Penguins dataset with all imputation methods. Optimal hyperparameters (shown below each method name) are obtained to minimize the NRMSE. kNNxKDE(m) stands for imputation performed with the mean of the returned samples from the kNNxKDE.

| *(hyperparams)* | $k$NN-Imputer 20 neighbors | MissForest 15 trees | MICE x | GAIN 1200 iterations | kNNxKDE default |
|---|---|---|---|---|---|
| **NRMSE** | 0.252 | 0.276 | 0.248 | 0.257 | 0.385 |
| **Log-Lik.** (Ref=$-2130$) | $-5683$ | $-4023$ | $-6309$ | $-5793$ | $-3008$ |

