# OpenReview forum: "Tabular data imputation: quality over quantity"
_NeurIPS.cc/2022/Conference — NeurIPS 2022 Submitted_

### Official Review · Reviewer_xjdE · 2022-07-10

**Rating:** 4
**Confidence:** 4
**Soundness:** 3 good
**Presentation:** 3 good
**Contribution:** 3 good

**Summary:**

This study presents a strategy for imputing missing values in tabular data in the presense of multi-modal distributions. The motivation is that existing imputation methods tend to give poor imputations for multi-modal datasets, the examples of which are described in the manuscript.


**Questions:**

Questions regarding baseline methods.
- Please provide the details of what you have tried for the baseline methods.
- Particually, I'm curious how you set the hyperparameters of GAIN, including the architecture and optimization. My experience says the performance of GAIN is not that bad (althuogh it's fairly sensitive to the hyperparemeter settings.)

Questiions regarding the proposed method.
- What would be the expected effert if the hyperparameter setting changes? How do we optimize it for the practical use?
- Will it work well on real-world datasets involving high-dimensionality?

**Limitations:**

Suggestions to improve experimental validation
- The authors used only two-dimensional toy datasets, which are not usual in real-world. The effectiveness of the proposed method should be evaluated on the datasets with various dimensionalities and also various sizes.
- The evaluation also need to be performed further under various missing rates as well as 20%.
- Among the baseline, for the k-NN imputation, a widely used way to improve the imputation is to use of median/mode over the observsed values in the k-NN, instead of using the average. This can give much better imputation for multi-modal datasets, including the toy datasets used in this study. This can be used as a baseline to be compared with the proposed method.
- An intuitive way to evaluate a imputation method is to use classificaiton/regression datasets. if it improves the performance for a classification/regression problems under the presense of missing values in input features.
- What would be the expected effect of the hyperparameters


**Strengths And Weaknesses:**

Strength
- I like the motivation of this study. Most existing imputation methods tend to give poor imputations under the multimodality. As authors mentioned, among plausable values for a missing value, using their average can be a very bad choice over simply picking one.
- I agree that the RMSE evaluation for the imputed values is not sensible in the evaluation of a missing value imputation method. the likelihood-based evaluation makes sense and seems better.

Weakness
- My main concern is that the scope of experimental validation is narrow and very limited. The details regarding this are listed in "Questions" section.
- It seems the hyperparameters baseline methods are not carefully tuned.
- Also, the effect of hyperparameters on the proposed method is not well-studied.

---

> ### Author Response · Authors · 2022-07-29
> **The hyperparameters of GAIN do not change much**
>
> Thank you for your time and valuable insight.
>
> The presentation of the kNNxKDE indeed does not make use of real-world datasets. The initial motivation behind this work is a multi-dimensional astrophysics dataset which imputation with GAIN or MissForest was very poor. After investigation, we wanted to highlight (using simple 2d datasets) that current standard imputation methods are not suited for multimodal datasets. We take good notes of your feedback and will work on the usefulness of the kNNxKDE by using real-world datasets. After revision, we will try to publish this work again in the future.
>
> --Answer to your questions—
>
> Q1 and Q2) Regarding GAIN, I have tried a wide range of hyperparameters for \alpha (the scaling factor between adversarial and RMSE loss for the generator) and h the hint-rate for the discriminator. I have even tried to change the architecture, in vain. My intuition is that the generative adversarial process has a hard time working on a cell-by-cell basis. I have contacted the authors of GAIN who could not provide with extra valuable insight. I would be happy to further exchange on this as I am very interested in making GAIN perform better.
>
> Q3) Typical effects in the change of the hyperparameters of the kNNxKDE are provided in Figures 5 and 6 of the paper. The Gaussian kernels bandwidth h plays the role of a smoothing factor, and the temperature \tau controls the width of each point’s neighborhood. We will further investigate the effects of h and \tau in real-world data imputation settings.
>
> Q4) We have already tried the kNNxKDE on a multi-dimensional astrophysics dataset, but this is the only real-world dataset as of now. We will use UCI datasets with classification/regression tasks to develop the kNNxKDE for practical purposes.
>
> Sincerely yours

---

### Official Review · Reviewer_WFqA · 2022-07-11

**Rating:** 3
**Confidence:** 4
**Soundness:** 1 poor
**Presentation:** 2 fair
**Contribution:** 1 poor

**Summary:**

The authors via this paper tried to highlight the shortcomings of the current imputation methods which tend to focus on minimizing the error between the unobserved and the imputed values. Authors propose a new qualitative approach with kNNxKDE algorithm i.e. a hybrid method using kNN and KDE capable of imputing missing features of a dataset with several modes. Authors use NRMSE to compare and evaluate the performance of their algorithm with other imputation methods.

**Questions:**

I understand RMSE cannot be used as an evaluation metric for all imputation methods, however, even NRMSE assumes smaller values for simulations underestimating the observations. Did the authors notice shortcomings (if any) by using NRMSE as an evaluation metric in their experiments?

**Limitations:**

Authors did a fairly good job in addressing the limitations of their study. However, these limitations are very critical in developing machine learning applications especially data imputation being the earliest stage in data preprocessing.

**Strengths And Weaknesses:**

The authors have rightly highlighted the shortcomings or the limits of root mean square error (RMSE) as an evaluation metric for the data imputation problems. Also, briefly discussed and emphasized the current available standard imputation methods and how they operate and key differences in each method. \
I believe the major weakness of this paper is lack of rigorous experimentation. Authors chose only a few simple datasets to run their experiments and compare with the current imputation methods, the study is only suitable for a narrow selection of datasets. In the real world, data is not always simple! Also, I recommend authors to expand the study to show the performance of their proposed algorithm with other evaluation metrics.

---

> ### Author Response · Authors · 2022-07-29
> **What other evaluation metrics could we use?**
>
> Thank you for your time and meaningful insight.
>
> Indeed, our work makes no use of real-world or high-dimensional data. We wanted to highlight on simple 2d datasets that current SOTA methods cannot perform good imputation. The initial motivation behind this work is a large astrophysics dataset which imputation could not be performed properly with GAIN or MissForest. We will work on the practical usefulness of the kNNxKDE by using real-world datasets. We will try to publish this work again in the future, after improvement.
>
> We have a question on our side: you suggest using “other evaluation metrics”, but what do you have in mind besides NRMSE or likelihood?
>
> --Answer to your question--
>
> We use the NRMSE instead of the RMSE because columns in a dataset can have different units and order of magnitude. By scaling all columns in the range [0, 1], we can compare the imputation quality between columns. We did not notice any particular shortcoming with the NRMSE, but some may happen in real-world datasets with outliers when we scale the dataset in the range [0, 1] for instance.
>
> Sincerely yours

---

### Official Review · Reviewer_4E75 · 2022-07-12

**Rating:** 3
**Confidence:** 4
**Soundness:** 1 poor
**Presentation:** 3 good
**Contribution:** 1 poor

**Summary:**

The paper addresses the limitations of current state-of-the-art (SOTA) data imputation algorithms at accurately modeling multimodal data.  The authors demonstrate this ineffectiveness using three toy synthetic datasets (2 dimensional with 500 samples each) and 4 SOTA algorithms.  An algorithm which combines the popular kNN-Imputer with kernel density estimation, called kNN-KDE, is presented and shown to well capture the three patterns in the toy datasets: linear, sinusoidal, and spherical.  The paper also makes a case for measuring imputation quality via the log-likelihood as opposed to the commonly used RMSE, while also admitting this is impractical on real-world datasets.

**Questions:**

"The best hyperparameters, presented in Table 1, are
used to impute each dataset one more time" <- This overfits each
imputation method and is not reflective of generalization
performance.  Given the data is synthetic and easily generated, why
not produce more samples and use cross-validation?

-Some discussion of the plausbility of the synthetic data seems
warranted.

**Limitations:**

The authors address several limitations of their work.  However, one important limitation is that the devised algorithm, kNN-KDE, is particularly designed to handle the three synthetic datasets presented (i.e., use of Gaussian kernels is ideal for spherical data).  This is an extremely limited representation of multimodal data, and non-spherical datasets could also be studied to determine the efficacy of kNN-KDE on a much wider variety of data.

**Strengths And Weaknesses:**

Originality:
Strengths: The presented data imputation algorithm is novel.  Updating the popular kNN-imputer to utilize a Gaussian kernel is interesting.

Weakness: In the main paper, the effectiveness of the proposed algorithm is not demonstrated either in a practical setting on real world data or on complicated/higher-dimensional synthetic datasets.  Furthermore, existing methods tackle the exact problem being tackled by kNN-KDE;  imputation methods using Gaussian copulas have been explicitly designed to accurately model multimodal data :
Zhao, Yuxuan, and Madeleine Udell. "Missing value imputation for mixed data via gaussian copula." Proceedings of the 26th ACM SIGKDD international conference on knowledge discovery & data mining. 2020.
--------------
Quality:
Weakness: Major weakness in the paper are inherent in the evaluation and the underlying motivation of the problem being tackled.
"real-world datasets do not come with a likelihood" <-
For the latter, the main paper does not consider real world datasets.  A major concern is the main problem being tackled--failures of existing SOTA imputation methods to model multimodal data--are motivated by synthetic datasets, rather than actual trends observed in real data.  It is also important to note that kNN-KDE is particularly designed to handle the three synthetic datasets presented (i.e., use of Gaussian kernels is ideal for spherical data), but this is an extremely limited representation of multimodal data and and may not reflect trends commonly encountered in ML data.  Furthermore, while kNNxKDE is described, the presented work is not an end-to-end solution: "Therefore, we are left with two options: either we use visual inspection and plots to assess the data imputation quality, or we optimize τ to minimizing the RMSE " <- At a minimum, the problem of how to run the algorithm on real data should be solved and presented in the paper.  Otherwise, researchers cannot put kNNxKDE into practice and the presented imputation framework is incomplete.

With regards to evaluation, while real data does not come with a log-likelihood, other evaluation exists with which to determine the
effectiveness of a data imputation approach (as is apparent in many of the works cited, e.g., Jager et al, 2021).  For instance, using a
UCI classification dataset with missing values, values could be imputed and downstream classification performance used to show the
performance of different imputation algorithms.

With regards to the advocacy of qualitative (log-likelihood) vs quantiative (RMSE) measures, the authors state:
"Ultimately, this work advocates for a qualitative approach of data imputation, rather than the current
quantitative one." <- As the authors have pointed out, this is impractical for real world data due to an inability to compute the log-likelihood.
--------------
Clarity:
The writing is clear and the work, as presented, is easy to parse and understand.
--------------
Significance:
Strengths: Data imputation is an important practical problem in machine learning.  As kNN-imputer is one of the most popular SOTA imputation algorithms, improvements to the algorithm may be widely appreciated and put to use by ML researchers.

Weakness:  As mentioned in the section on quality, practical evaluation of kNN-KDE remains lacking, so that the performance and significance of the algorithm on commonly encountered real-world datasets is unknown.

---

> ### Author Response · Authors · 2022-07-29
> **We would happily share the synthetic data generation code**
>
> Thank you for your time and valuable feedback.
>
> We understand that our method lacks real-world applications and practical considerations. The motivation behind this work is a large astrophysics dataset which imputation could not be performed properly with GAIN or MissForest.
>
> We will improve the usefulness of our method by making use of real-world datasets and designing an end-to-end solution for practical data imputation. We will try to publish this work again in the future, after improvement.
>
> -- Answer to your questions --
>
> Q1) You are correct in saying that re-using the same hyperparameter values is overfitting. We will generate new data again for testing from now on.
>
> Q2) We can provide the code for the synthetic data as we generated them with a seed. They are simple 2d data with multimodal structures.
>
> Sincerely yours

---

> > ### Comment · Reviewer_4E75 · 2022-08-09
> > **Response to the authors**
> >
> > I thank the authors for their response, and wish them luck on future iterations of the paper.

---

### Meta-Review · Area_Chair_nn6E · 2022-08-29

**Recommendation:** Reject
**Confidence:** Certain

**Metareview:**

This paper makes an interesting observation that imputation
error metrics may work poorly for multimodal data, and proposes a creative solution.
However, empirical evaluation of the idea on real datasets,
and comparison to other proposals for imputation with multimodal data, are lacking. I would encourage the authors to take the reviewers' suggestions seriously and resubmit to a future conference.

**Award:**

No

---

### Decision · Program_Chairs · 2022-09-14

Reject